# Invisible Silver Nanomesh Skin Electrode via Mechanical Press Welding

**DOI:** 10.3390/nano10040633

**Published:** 2020-03-28

**Authors:** Ji Soo Oh, Jong Sik Oh, Geun Young Yeom

**Affiliations:** 1School of Advanced Materials Science and Engineering, Sungkyunkwan University, Suwon 16419, Korea; jsoh9689@skku.edu (J.S.O.); ojs2k@skku.edu (J.S.O.); 2SKKU Advanced Institute of Nano Technology (SAINT), Sungkyunkwan University, Suwon 16419, Korea

**Keywords:** silver nanowire, polydimethylsiloxane (PDMS), transparent conductive film, wearable conductive film, welding

## Abstract

Silver nanowire (AgNW) has been studied as an important material for next-generation wearable devices due to its high flexibility, high electrical conductivity and high optical transmittance. However, the inherently high surface roughness of AgNWs and low adhesion to the substrate still need to be resolved for various device applications. In this study, an embedded two-dimensional (2D) Ag nanomesh was fabricated by mechanical press welding of AgNW networks with a three-dimensional (3D) fabric shape into a nanomesh shape, and by embedding the Ag nanomesh in a flexible substrate. The effect of the embedded AgNWs on the physical and electrical properties of a flexible transparent electrode was investigated. By forming embedded nanomesh-type AgNWs from AgNW networks, improvements in physical and electrical properties, such as a 43% decrease in haziness, 63% decrease in sheet resistance, and 26% increase in flexibility, as well as improved adhesion to the substrate and low surface roughness, were observed.

## 1. Introduction

Recently, as the need for wearable devices has increased, so has the active research on flexible and stretchable electronics. For wearable devices, the electrodes have to be twistable, conformable, and deformable to fit the movement of the human body, and possess flexible and stretchable characteristics. They also need to exhibit low surface roughness, low haze characteristics, and high transmittance when the devices are used for displays [1,2,3,4,5,6,7,8,9].

To meet the requirements of the transparent electrodes for flexible and stretchable electronics, many researchers have studied new materials, such as graphene [10,11,12,13], carbon nanotubes (CNTs) [14,15,16,17], conductive polymers [18,19,20], and silver nanowires (AgNWs) [21,22,23,24,25]. Currently, among the investigated materials, AgNWs exhibit very high transmittance and high conductivity, and can be easily and inexpensively fabricated as transparent electrodes using various methods, such as spray coating, brush coating, spin coating, electro-spinning, jetting, etc. Nevertheless, the AgNWs show high surface roughness and high haziness caused by tangling among the AgNWs in the AgNW network, as well as a low adhesion between the AgNWs and the substrate [26,27,28,29,30].

Various planarization methods for the AgNW network have been widely studied to solve the above problems, such as (i) coating of a conductive polymer, e.g., poly(3,4-ethylendeioxythiophene):poly(styrenesulfonate) PEDOT:PSS onto the AgNW network [31,32,33,34]; (ii) embedding the AgNWs in a polymer thin film [23,25,30,35,36]; (iii) application of a physical, chemical force to the AgNW network, etc. [37,38,39,40,41]. These processes reduce surface roughness and improve adhesion. When a conductive polymer was coated on AgNWs, even though the roughness could be significantly reduced, due to the physical properties (e.g., the bandgap) of the conductive polymer, the optical transparency was decreased. A transparent electrode fabricated by embedding AgNWs in a polymer film showed ultra-smooth roughness characteristics. However, the surface resistance was not uniform because there was a region where the AgNWs were not exposed on the surface. Etching the embedded AgNW surface to increase the conducting surface area by exposing the AgNWs again increased the surface roughness. In the case of planarization by pressurizing the AgNW network coated on the substrate, there was a limit to the surface planarization due to the AgNW thickness itself.

In this study, we propose an ultra-thin embedded AgNW nanomesh electrode to improve upon the physical and electrical properties of previous transparent electrodes. The initial AgNW network, which was a three-dimensional (3D) structure of wires woven like fabric, was modified to produce a two-dimensional (2D) AgNW nanomesh. This was achieved by pressurizing the AgNW network and by embedding it in a transparent and flexible polymer to be used as a substrate. After forming the ultra-thin embedded AgNW nanomesh electrode, its effects on the haziness, contact resistance, stretchability, and adhesion to the substrate were investigated considering a potential application to transparent electrodes for wearable electronics.

## 2. Materials and Methods 

### 2.1. Materials

Poylmethyl methacrylate (950 PMMA C4 resist, Microchem, Westborough, MA, USA), the solution of the silver nanowire (diluted with 8–12 mL of isopropyl alcohol to 0.05 wt%, 25 ± 5 μm long and 25 ± 5 μm in diameter, NANOPYXIS, Korea), polydimethylsiloxane (PDMS; Sylgard 184, Dow Corning, Midland, MI, USA), polyimide film (PI; SKCKOLONPI, Korea) and Tert-Butyl alcohol (> 99.5%, Sigma-Aldrich, St. Louis, MO, USA) were used as received.

### 2.2. Welding Process

The AgNW junction was welded, and this was followed by the application of physical force ranging between 500 and 2000 N/cm^2^ using lab-made mechanically pressurizing equipment. 

### 2.3. Ultra-Thin Film Preparation

The PMMA was spin-coated on the SiO_2_ at 4000 rpm for 40 sec, then baked on a hotplate at 140 ℃ for 5 min. The solution of AgNWs was spray-coated onto the PMMA/SiO_2_ substrate using spray-coating equipment (15 psi of N_2_ gas, air spray gun; GP-35; SPARMAX, Germany) and a heated X-Y stage (heated to ~50 ℃). For embedding in the flexible and stretchable polymer substrate, liquid PDMS was prepared by mixing the base and a curing agent with Tert-Butyl alcohol at various ratios. The liquid PDMS was spin-coated for 30 sec at 5000 rpm, and thermal curing occurred at 60 °C for 4 hours in the oven. The AgNW-embedded PDMS/PMMA/SiO_2_ was immersed in 90 °C DI water, enabling it to be peeled off from the SiO_2_ by itself, followed by immersion in acetone for some time to remove the PMMA layer in sequence. 

### 2.4. Characterization

The surface properties of the AgNW nanomesh embedded in the PDMS film were investigated using field emission scanning electron microscopy (FE-SEM; S-4700, Hitachi, Japan) and atomic force microscopy (AFM; XE100, PSIA, Korea). Changes in the AgNW network morphology were measured by the energy dispersive X-ray (EDX; Sirion, FEI, Netherlands) and transmission electron microscopy (HRTEM; JEM-2100F, JEOL, Japan) before and after pressurized welding. The optical transmittance was measured by ultraviolet-visible (UV-Vis) spectroscopy (UV-3600; Shimadzu, Japan). The haze values were measured with a haze meter (Haze-Gard I; BYK-Gardner, Germany). The sheet resistance and mechanical integrity were measured using a four-point probe method and a lab-made bending/stretching test machine, respectively.

## 3. Results and Discussion

Figure 1 exhibits the procedure for fabricating the ultra-thin embedded AgNW nanomesh electrode. Polymethyl methacrylate (PMMA) was used as the adhesive and detachable base layer for the deposition of the AgNW electrode on the SiO_2_ substrate. Before the spray coating of the AgNWs on the SiO_2_ substrate, the PMMA was first spin-coated on the SiO_2_ substrate and subsequently baked on the hotplate to remove the solvent (Figure 1a). A solution of AgNWs was then spray-coated on the PMMA-coated substrate using lab-made spray equipment (Figure 1b). The number of AgNW spray coatings was varied to obtain an AgNW network with differing sheet resistances. Then, the 3D AgNW network formed on the substrate was mechanically pressurized to weld the AgNW junctions and to form a 2D nanomesh by placing a graphite sheet-covered SiO_2_ plate on the AgNW network/PMMA/SiO_2_ substrate. This was followed by the application of various physical forces (Figure 1c). The graphite sheet was placed between the SiO_2_ plate and the AgNW network/PMMA/SiO_2_ substrate to prevent damage to the substrate and to uniformly disperse the force. By removing the SiO_2_ plate and graphite sheet, a 2D Ag nanomesh was obtained (Figure 1d). Liquid polydimethylsiloxane (PDMS) was prepared by mixing the base and curing agent to embed the AgNW nanomesh electrodes in the flexible and stretchable polymer substrate. After all the air bubbles in the liquid PDMS were cleared, the liquid PDMS was spin-coated on the AgNW/PMMA/SiO_2_ substrate, and thermal curing followed to form the cross-linked solid PDMS (Figure 1e). The thermally-cured AgNW-embedded PDMS/PMMA/SiO_2_ was immersed in hot deionized (DI) water, enabling it to be peeled off by itself from the SiO_2_ substrate, followed by immersion of the AgNW-embedded PDMS/PMMA in acetone for some time to remove the PMMA layer on the AgNW-embedded PDMS. The AgNW-embedded PMDS was subsequently rinsed with alcohol and DI water to form a free-standing AgNW-embedded PDMS film (Figure 1f,g).

Figure 2 shows the change in optical transmittance (%), haziness (%), and sheet resistance (Rs, Ω/sq) for an AgNW network on a PMMA/SiO_2_ substrate, measured as a function of the number of AgNW spray coatings before mechanical welding by pressurizing (Figure 2a), and an AgNW nanomesh on a PMMA/SiO_2_ substrate, measured as a function of the pressure for mechanical welding of the AgNW network in Figure 2a with 20 sprayings (Figure 2b). As shown in Figure 2a, the increase in the number of AgNW sprayings from 5 to 20 decreased the sheet resistance from 5 kΩ/sq to 12.3 Ω/sq, while increasing the haziness from 2.52% to 11.3%, and decreasing the optical transmittance from 96.85% to 88.81%. Therefore, even though the increased number of AgNWs on the substrate improved the conductivity of the electrode, its optical properties as a transparent electrode, such as transmittance and haziness, were degraded.

After spraying 20 times, the AgNW network in Figure 2a was pressurized for mechanical welding, as shown in Figure 2b. The increased vertical pressure to 2000 N/cm^2^ not only decreased the sheet resistance to 8.4 Ω/sq and haziness to 7.12%, but also increased the optical transmittance to 90.17%. Therefore, through the pressurized mechanical welding, all the properties of the transparent electrode were improved. Similar mechanical welding experiments were performed for spraying 5, 10, and 15 times. Similar changes in optical transmittance, sheet resistance, and haziness were observed, as shown in Appendix A. Notably, the haziness was decreased most significantly with 20 sprayings (due to the highest haziness), and the sheet resistance was reduced most significantly with 5 sprayings (due to the highest initial sheet resistance).

To understand the reason for the improvements in the optical transmittance, sheet resistance, and haziness of the transparent electrode, we observed the change in the shape of the AgNW network on the PMMA/SiO_2_ substrate with increasing vertical pressure. Figure 3 shows the SEM images of AgNW networks with 20 sprayings before pressurizing (Figure 3a), and after pressurizing with vertical pressures of 500, 875, 1250, 1625, and 2000 N/cm^2^ (Figure 3b–f). When AgNWs were spray-coated on the substrate, as shown in Figure 3a, the AgNWs were stacked randomly, and the contact area between the AgNWs at the junction was small or no contact was formed at the junction. Therefore, a 3D AgNW network, with different heights at the junction, was formed on the substrate surface. As the AgNW network was pressurized, as shown in Figure 3b–f, with the increase in the vertical pressure from 500 to 2000 N/cm^2^, more contact area was formed at the AgNW junctions and, especially when the pressure was higher than 1625 N/cm^2^, the 3D AgNW network changed into a 2D Ag nanomesh. 

Figure 4 shows the atomic force microscopy (AFM) image and the AFM height differences between single AgNW and double AgNW junctions in the AgNW network after spray coating the AgNW 20 times (Figure 4a), and the AgNW nanomesh after pressurizing with 2000 N/cm^2^ (Figure 4b). As shown in Figure 4a, the height of the double AgNW junction was about twice that of the single AgNW due to the simple stacking of the AgNWs at the junction. Also, the surface roughness values of the AgNW network were 54.8 nm for the RMS surface roughness (R_a_) and 130 nm for the peak-to-valley surface roughness (R_p–v_). In the case of the Ag nanomesh obtained after pressurizing with 2000 N/cm^2^, as shown in Figure 4b, the height of the double AgNW junction was similar to that of the single AgNW due to the complete mechanical welding at the junction. Also, the surface roughness values of the 2D Ag nanomesh were 28.8 nm for R_a_ and 30 nm for R_p–v_, values similar to the diameter of the AgNW. Therefore, a very smooth surface roughness was observed after the mechanical welding of the AgNW network at 2000 N/cm^2^.

Using cross-sectional TEM, the change in the shape of the AgNW network after the pressurizing was further investigated. Figure 5a shows the AgNW network after 20 spray coatings, and Figure 5b shows the Ag nanomesh formed after pressurizing the AgNW network in Figure 5a with 2000 N/cm^2^. As shown in Figure 5a,b, when the AgNWs were spray-coated, the random 3D stacking of AgNWs was observed but, after the pressurizing of the AgNW network, a planar 2D AgNW was observed. Figure 5c shows the energy dispersive X-ray EDX data of the Ag nanomesh on the SiO_2_ substrate from Figure 5b, and also shows a planar Ag nanomesh formed on the substrate by the pressurizing of the AgNW network. Platium (Pt) was coated on the Ag nanomesh to prepare the cross-sectional TEM of the Ag nanomesh on the SiO_2_ substrate. Figure 6 illustrates the reason for the change in the optical properties when forming a 2D Ag nanomesh from a 3D AgNW network. When the AgNWs on the substrate were in a 3D network, the contact area between the AgNWs at the junction was small and, due to the thickness variation of nanowires, the light was easily scattered at the junction.

However, for the 2D Ag nanomesh, the junction was fused entirely, and the scatter angle at the AgNW junction was smaller than that of the AgNW network, as shown in Figure 6a,b. Therefore, it is believed that, after the formation of a 2D Ag nanomesh, due to the decreased light scattering and lower angle scattering, the optical transmittance and haziness [42,43,44,45] are improved, and there is a decreased sheet resistance from the fusing of the contact junction area.

On the Ag nanomesh/PMMA/SiO_2_, a PDMS solution, diluted using Butanol (Tert-Butyl alcohol) from 1:0 (PDMS:Butanol) to 1:30, was applied by spin coating, and a rectangular-shaped polyimide (PI) film was attached around the substrate as a frame for easy handling of the electrode, and then cured at 120 ℃ for 10 min (Figure 1e). After the removal of the PMMA/SiO_2_ substrate, a free-standing AgNW-embedded PDMS film framed by PI was obtained and, after it was attached to human skin, as shown in Figure 1h, an invisible Ag nanomesh skin electrode was formed after the removal of the PI frame. For the fabrication of an invisible skin electrode, the thickness of the PDMS embedded with the Ag nanomesh is critical and, if the PDMS thickness is not thin enough, the electrode is easily peeled off from human skin during skin movement. The Butanol ratio was increased from 1:0 (PDMS:Butanol) to 1:30, and the spin speed was also increased from 1000 to 6000 rpm. As shown in Figure 7a, a final thickness of Ag nanomesh-embedded PDMS from ~35 μm (1000 rpm and 1:0) to ~140 nm (6000 rpm and 1:30) could be obtained. Free-standing transparent electrodes were prepared by spin coating a PDMS solution (1:30 PDMS:Butanol) on the 3D AgNW network/PMMA/SiO_2_ and on the 2D Ag nanomesh/PMMA/SiO_2_ substrate in Figure 4, and by peeling off the AgNW-embedded PDMS/PMMA. After removing the PMMA, the area of the AgNWs exposed on the surface of the PMDS was examined for both the 3D AgNW network containing PDMS, and the 2D Ag nanomesh containing PDMS.

Figure 7b,c show the SEM images of the AgNWs on the PDMS surface for the 3D AgNW network and the 2D Ag nanomesh, respectively. Inset figures are the images of the Ag area on the PDMS surface for calculating the total Ag area ratio using a surface area measurement software (Image J; NIH). The 2D Ag nanomesh-embedded PDMS showed a larger Ag electrode area on the PDMS surface, as shown in the SEM images. The calculation also showed that the ratio of the AgNW electrode to the PDMS area was 24.5%, compared to 12.8% for the 3D AgNW network-embedded PDMS, showing that more current flows for the 2D Ag nanomesh-embedded PDMS. 

The effect of the flexible/stretch stability of the AgNW-embedded PDMS electrodes on the change in resistance was investigated, and the results are shown in Figure 8. For the AgNW network and the Ag nanomesh, electrodes (7 cm × 5 cm × 140 nm) were fabricated and, after attaching to a PDMS support, the bending/stretching stability was investigated. After forming Ag paste stripes (Appendix A), the change in resistance as a function of the mechanical stability was measured using a four-point probe method and a lab-made bending/stretching test machine. Figure 8a shows the change in sheet resistance of the AgNW-embedded PDMS electrodes, measured as a function of the bending cycle. The mechanical integrity of the electrodes was measured using a lab-made bending test system bended to a 3 mm radius of curvature. In the case of a 3D AgNW network-embedded PDMS electrode, a resistance increase of 873.82% was observed after 10,000 cycles, despite being embedded in PDMS. On the other hand, although there were slight differences depending on the pressurized values, the 2D Ag nanomesh-embedded PDMS electrodes showed excellent durability in the bending test. In particular, in the case of electrodes fabricated by applying 2000 N/cm^2^, a resistance increase of 0.25% after 10,000 cycles was observed. Figure 8b shows the optical images of the invisible Ag nanomesh skin electrode obtained by attaching 2D Ag nanomesh-embedded PDMS to human skin. The 140 nm thick Ag nanomesh skin electrode attached well to the human skin and was not detectable by the human eye, as shown in the optical images. As shown in Figure 8c, for the AgNW network-embedded PDMS electrode, the resistance did not change until the strain was ~20%; however, when the strain increased even further, the resistance increased significantly. When the strain was higher than 40%, the resistance was almost infinite due to the cut-off of the AgNW network in the PDMS. In the case of the Ag nanomesh-embedded PDMS electrode, no significant change in resistance was observed until 80% stretch strain was reached, and thereafter, a significant change in resistance was seen.. In fact, for the AgNW network-embedded PDMS, the resistance changed from 7.9 Ω/sq at no strain to 308 Ω/sq at 40% (~3800% increase), while for the Ag nanomesh-embedded PDMS, it changed from 6.2 Ω/sq at no strain to 9.7 Ω/sq at 40% (~56% increase) (Appendix A). The electrical uniformity of the electrodes fabricated in this paper was performed (Appendix A), furthermore, stretching cyclic tests were performed under the conditions of 20%, 40%, and 60% stretch strain, and the results are shown in Figure 8d. Stretch and release were repeated for 1000 cycles at various conditions to check the change in resistance: at 20% (R_0_ = 7.62 Ω/sq), the value at stretch was 9.45 Ω/sq, and the value at release was 7.79 Ω/sq; at 40% (R_0_ = 6.89 Ω/sq), values of 11.11 Ω/sq at stretch and 7.10 Ω/sq at release were obtained, and at 60%, 24.14 Ω/sq at stretch and 8.01 Ω/sq at release were noted. Thus, the initial resistance was observed for the strain of 20% and 40% after the release and, for the strain of 60%, even though a slightly higher resistance was observed after the release, the increase in the resistance was not significantly large. Therefore, a significant improvement in maintaining resistance during stretching could be observed for the Ag nanomesh-embedded PDMS.

## 4. Conclusions

In this study, by forming a planar Ag nanomesh through the mechanical welding of a spray-coated AgNW network and by embedding the planar Ag nanomesh in an extremely thin PDMS film, an invisible skin electrode, which is durable and has low resistance, very high optical transmittance, and very low haziness, could be fabricated. The electrodes formed by the spray-coated AgNW network showed degraded optical properties, with decreased optical transmittance and increased haziness, but improved electrical properties, with the sheet resistance being decreased with an increase in the number of spin coatings of the AgNWs. By forming a 2D Ag nanomesh through pressurizing the spray-coated AgNW network, one observed not only an improvement in optical properties, with increased optical transmittance (from 88.8% to 90.2%) and decreased haziness (from 11.3% to 7.1%), but also an improvement in the electrical properties, with decreased sheet resistance (from 12.3 Ω/sq to 8.4 Ω/sq), was obtained by pressurizing the AgNW network up to 2000 N/cm^2^. The improvement in electrical properties by forming a 2D Ag nanomesh from a 3D AgNW network was related to the fusing of the AgNW junctions; the improvement in the optical properties was related to the decrease in light scattering and in light scatter angle by forming a 2D Ag nanomesh. The invisible skin electrode was fabricated by embedding the Ag nanomesh in extremely thin PDMS (by forming a 140 nm~ thick Ag nanomesh-embedded PDMS film), and the invisible Ag nanomesh skin electrode was not only well-attached to human skin but also showed no significant change in resistance up to 80% stretch strain. It is believed that the Ag nanomesh-embedded PDMS can be applied to various next-generation devices requiring invisible skin electrodes. 

## Figures and Tables

**Figure 1 nanomaterials-10-00633-f001:**
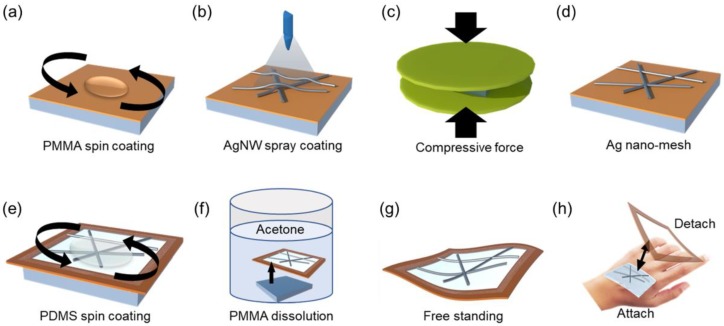
Schematic illustration of the fabrication process for the embedded silver nanowire (AgNW) nanomesh transparent skin electrode. (**a**) The PMMA was first spin-coated on the SiO_2_ substrate and baked on the hotplate. (**b**) A solution of AgNWs was spray-coated on the PMMA-coated substrate using spray equipment. (**c**,**d**) Mechanically pressurized to weld the AgNW junctions, then obtained the 2D Ag nanomesh. (**e**) Liquid PDMS was spin-coated on AgNWs/PMMA/SiO_2_ substrate, and thermal curing. (**f**) Peel-off by itself from the SiO_2_ substrate in hot DI water, followed by immersion of the AgNW-embedded PDMS/PMMA in acetone for remove the PMMA layer. (**g**,**h**) Subsequently rinsed with alcohol and DI water to form a free-standing AgNW-embedded PDMS film.

**Figure 2 nanomaterials-10-00633-f002:**
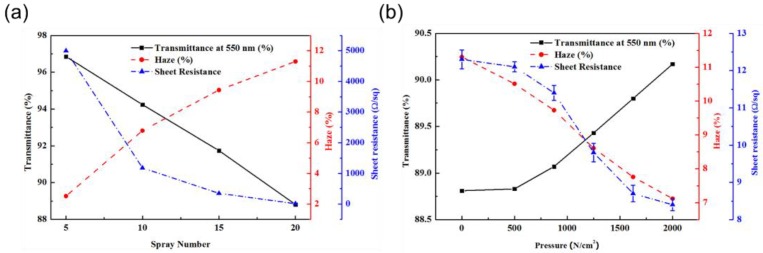
Change in optical transmittance at 550 nm, haze, and sheet resistance: (**a**) as a function of the number of AgNW spray coatings before mechanical welding by pressurizing the AgNW network on the polymethyl methacrylate (PMMA)/SiO_2_ substrate measured; (**b**) as a function of the pressure for the mechanical welding of the AgNW in (**a**) with 20 sprayings.

**Figure 3 nanomaterials-10-00633-f003:**
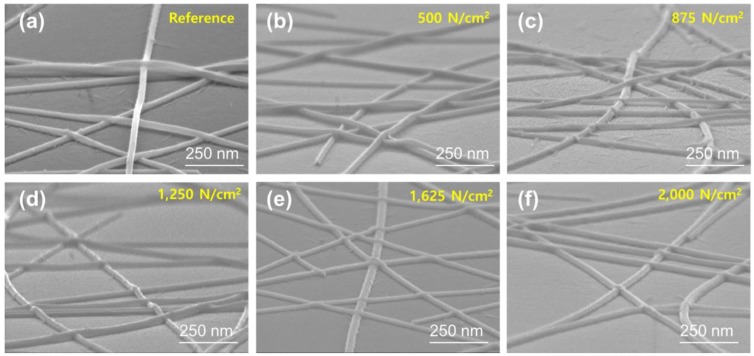
SEM images of the AgNW networks with 20 sprayings before pressurizing (**a**) and after pressurizing with the vertical pressures of (**b**) 500; (**c**) 875; (**d**) 1250; (**e**) 1625; and (**f**) 2000 N/cm^2.^.

**Figure 4 nanomaterials-10-00633-f004:**
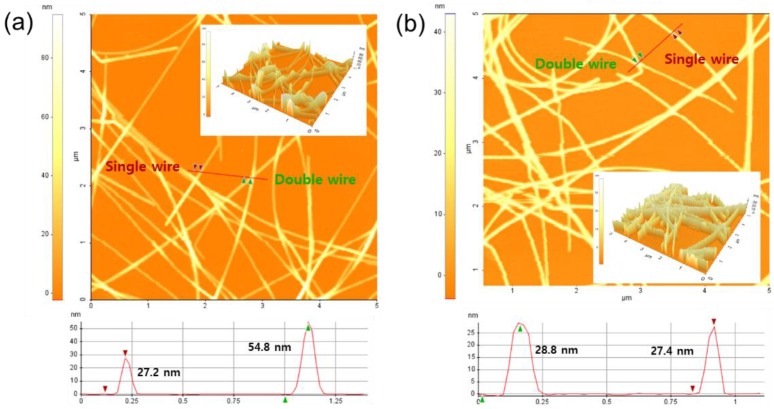
Atomic force microscopy (AFM) image and the AFM height differences between the single AgNW junction and the double AgNW junction (**a**) with 20 AgNW spray coatings (before pressurizing) and (**b**) the AgNW nanomesh after pressurizing with 2000 N/cm^2^.

**Figure 5 nanomaterials-10-00633-f005:**
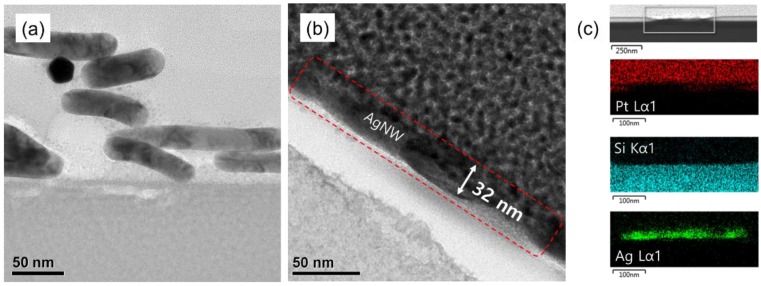
Cross-sectional TEM images of the AgNW network (**a**) before and (**b**) after pressurizing with 2000 N/cm^2^; (**c**) EDX data of the Ag nanomesh on the SiO_2_ substrate after pressurizing. A Pt coating was applied after the Ag nanomesh was formed on the SiO_2_ surface to observe the cross-sectional TEM.

**Figure 6 nanomaterials-10-00633-f006:**
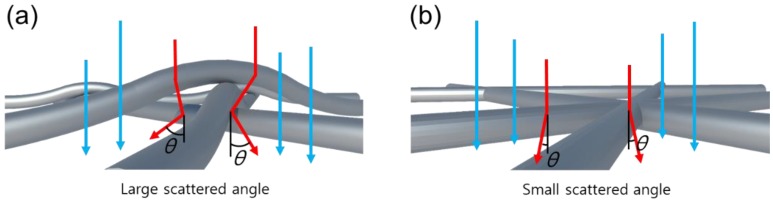
The illustration explains the reason for the change in the optical properties when a 2D Ag nanomesh is formed from a 3D AgNW network. (**a**) large scattered angle when it shaped 3D AgNW network, (**b**) small scattered angle when it shaped 2D Ag nanomesh

**Figure 7 nanomaterials-10-00633-f007:**
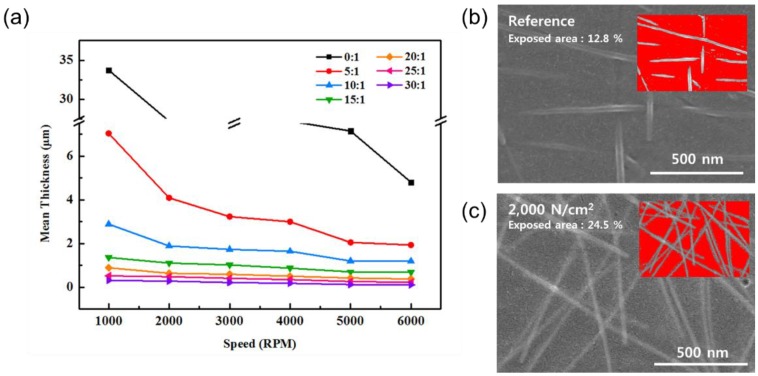
(**a**) The thickness of the AgNW-embedded polydimethylsiloxane (PDMS) film using various weight ratios (from 1:0 to 1:30) of PDMS:Butanol as a function of the spin speed. SEM images of AgNWs exposed to the outside of PDMS for (**b**) the 3D AgNW network and (**c**) the 2D Ag nanomesh (pressurized at 2000 N/cm^2^).

**Figure 8 nanomaterials-10-00633-f008:**
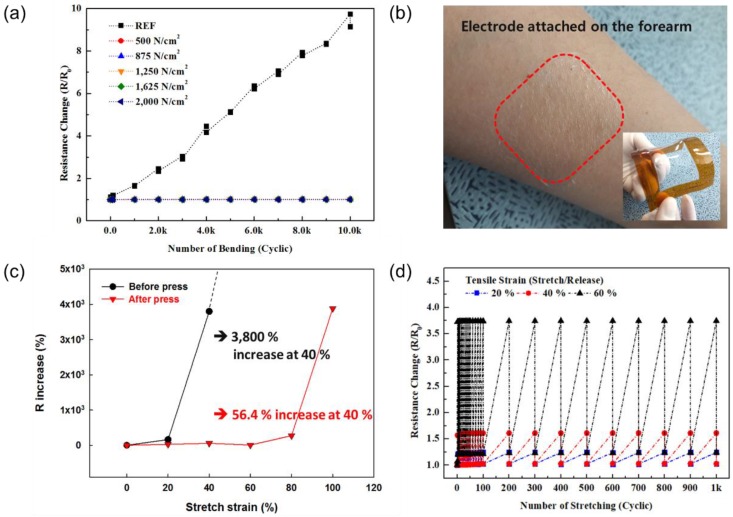
Flexibility and stretchability of the AgNW electrode embedded in PDMS where (**a**) shows the change in sheet resistance during the cyclic bending (radius = 3 mm) for various pressure conditions (detailed resistance changes for each condition can be found in Appendix A); (**b**) an optical image of an Ag nanomesh skin electrode, invisible to the naked eye, obtained by attaching 2D Ag nanomesh-embedded PDMS to human skin; (**c**) the change in sheet resistance of the AgNW-embedded PDMS before (with the 3D AgNW network) and after (with the 2D Ag nanomesh) pressurizing at 2000 N/cm^2^, measured as a function of stretch strain; (**d**) the effect of repeated stretching and releasing on the resistance (stretch/release cycles of tensile strain = 20%, 40%, and 60%).

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
