# Peer review of "Invisible Silver Nanomesh Skin Electrode via Mechanical Press Welding"

_nanomaterials, 2020, doi:10.3390/nano10040633_

Round 1

Reviewer 1 Report

I suggest to authors to perform electrical resistance measurements of planar Ag nano-mesh electrodes. However, the work quality is optimum also at this stage and can be accepted as is.

Author Response

We received the reviewer comments from Nanomaterials. List of comments & replies to reviewer critical comments have written as followed. We sincerely thank you for your helpful comments.

As the reviewer advised, we checked and corrected the English grammar and spelling.

Thank you for your consideration.

Reviewer 2 Report

This work reports an embedded two-dimensional (2D) Ag nano-mesh that fabricated by mechanical press welding of silver nanowire networks that make a three-dimensional (3D) fabric shape into a nano-mesh flexible substrate. Various techniques have been used for characterization. Some results have been obtained, leading to being interesting for the readership in the area of wearable devices.  And thus, this manuscript may be acceptable, however, subject to addressing the following major issues:

  1. How to tell the essential difference between "optical transmittance" and "haziness"?
  2. What happens for the electrical conductivity if a vertical pressure is increased to higher than 2000 N/cm2?
  3. For long-duration use, the stretch/bending stability should be tested over 100 cycles, and details to be provided in the revision.
  4. The authors need to confirm the electrical distribution uniformity at the center and edge of the transparent electrodes, using suitable techniques.
  5. What's the optimized thickness of PDMS? Is 140nm thick enough for the electrodes?
  6. The error bar for the sheet resistance (Fig. 2b) needs to add in the revision.
  7. The English writing should be improved because of a number of typos/grammar errors, for example, (1) Line 48, "it it showed" should be "it showed"; (2) Line 54, "a ultra-thin" should be "an ultra-thin"; (3) Line 69, "a solution of " should be "A solution of"; (4) Line 106, cm2 should be superscript, and so on.

Author Response

We received the reviewer comments from Nanomaterials. List of comments & replies to reviewer critical comments have written as followed. We sincerely thank you for your helpful comments.

As the reviewer advised, we corrected the English typos/grammar errors.

Thank you for your consideration.

Reviewer 3 Report

The paper "Invisible Silver Nano-mesh Skin Electrode via 2 Mechanical Press Welding" describes a method to obtain a Ag nanowire nanomesh from a Ag nanowire network, by applying mechanical pressure on the network. The obtained nanomesh shows improved electrical and optical properties, which is supported by different characterization methods. 

In general, the results are interesting and the paper is scientifically sound, I thus recommend publication on Nanomaterials, provided that the following points are addressed.

  • the authors should specify the wavelength at which transmittance measurements are taken
  • the paragraph between lines 186-196 is not very clear to me. In particular, it is not clear what the authors have done, the purpose of it and the meaning of exposed area. I would recommend rewriting the paragraph making such points clearer

Author Response

(The authors gave the same response as above.)

Round 2

Reviewer 2 Report

The revision is satisfactory for publishing at the current status, although the English writing still needs further polishing.